# Influence of renal function and daptomycin dose on clinical effectiveness and adverse events in Japanese pediatric patients: A multicenter retrospective observational study

Chihiro Shiraishi[1,2], Hideo Kato[1,2*], Jun Hirai[3], Naoya Nishiyama[4], Takayuki Inagaki[5], Nobuyuki Ashizawa[6], Masashi Nakamatsu[4], Kazuko Yamamoto[4], Yasuhiro Miyagawa[7], Hiroyuki Honda[6], Satoshi Kakiuchi[6], Ayumi Fujita[6], Masato Tashiro[6], Takeshi Tanaka[6], Koichi Izumikawa[6], Mao Hagihara[8,9], Nobuhiro Asai[8], Nobuaki Mori[8], Hiroshige Mikamo[8], Takuya Iwamoto[1,2]

1 Department of Pharmacy, Mie University Hospital, Mie, Japan, 2 Department of Clinical Pharmaceutics, Division of Clinical Medical Science, Mie University Graduate School of Medicine, Mie, Japan, 3 Department of Infection Control Department, Nippon Medical School Chiba Hokusoh Hospital, Chiba, Japan, 4 First Department of Internal Medicine, Division of Infectious, Respiratory, and Digestive Medicine, University of the Ryukyus Graduate School of Medicine, Okinawa, Japan, 5 Division of Pharmaceutical Sciences I, Faculty of Pharmacy, Meijo University, Aichi, Japan, 6 Infection Control and Education Center, Nagasaki University Hospital, Nagasaki, Japan, 7 Department of Hospital Pharmacy, Nagoya University Hospital, Aichi, Japan, 8 Department of Clinical Infectious Diseases, Aichi Medical University, Aichi, Japan, 9 Department of Molecular Epidemiology and Biomedical Sciences, Aichi Medical University, Aichi, Japan

* hkato59@med.mie-u.ac.jp

## Abstract

### Background

Data on the clinical effectiveness and adverse events of daptomycin in pediatric patients, considering dosage and renal function, are limited.

### Aim

In this study, we aimed to investigate the clinical effectiveness of daptomycin, incidence of related adverse events, and associations between clinical outcomes and risk factors, including dosage and renal function, in pediatric patients.

### Methods

Medical records of pediatric patients treated with daptomycin between September 2011 and December 2022, were retrospectively reviewed. Clinical effectiveness was categorized as cure, improvement, failure, or non-evaluable. The clinical success rate was defined as the sum of cured and improved cases. We classified doses based on the approved ranges: underdose (>1 mg/kg less than the approved dose), adequate dose (approved dose ±1 mg/kg), and overdose (>1 mg/kg more than the approved

**Data availability statement:** All relevant data are within the paper and its Supporting Information files.

**Funding:** The work was supported by Morinomiyako Medical Research Foundation (J23032A010). The funders had no role in study design, data collection and analysis, decision to publish, or preparation of the manuscript.

**Competing interests:** The authors have declared that no competing interests exist.

dose). Obesity was defined as a body mass index $\geq 30\,kg/m^2$, with adjusted body weight used for dosing in these patients.

## Findings

We enrolled 54 patients, achieving a clinical success rate of 91%. Four (7%) patients died, particularly those with microbiological failure ($P = 0.004$) and underdosing ($P < 0.001$). The underdose group comprised a higher proportion of younger patients ($P = 0.020$). Additionally, seven (13%) patients experienced daptomycin-related adverse events, including creatine phosphokinase elevation, eosinophilic pneumonia, rhabdomyolysis, liver failure, and drug fever, occurring regardless of renal function. Five patients received an approved dose, while two received an overdose.

## Conclusion

Adequate daptomycin dosing improved clinical effectiveness and the mortality rate. However, continuous monitoring of adverse events remains critical for patients receiving the approved dose, including those with a normal renal function.

## Introduction

Daptomycin is a lipopeptide antibiotic with broad-spectrum activity against gram-positive bacteria, including staphylococci, enterococci, and streptococci [1]. This antibiotic is recommended as an alternative for treating methicillin-resistant *Staphylococcus aureus* (MRSA) with high vancomycin minimum inhibitory concentrations (MIC) and treatment failure [2]. Consequently, its use in such cases has been increasing [3]. Daptomycin has been reported to improve clinical symptoms and 30-day mortality rates, demonstrating effectiveness comparable with that of vancomycin in treating bacteremia, complicated skin and soft tissue infection (cSSTI), and endocarditis caused by MRSA in adult patients [4–6]. Additionally, the cost of daptomycin treatment may also be comparable with that of vancomycin, considering that vancomycin therapy requires area under the curve (AUC) and trough-based therapeutic drug monitoring [7,8]. Most guidelines recommend daptomycin for treating bacteremia or cSSTI caused by gram-positive bacteria in adult patients [9–11].

In many countries, including Japan, daptomycin has been approved for treating pediatric patients aged 1–17 years with bacteremia and cSSTI [12,13]. The proposed daptomycin dosage for pediatric patients is based on age, considering the wide range of physical development processes that occur from the newborn stage to adolescence [14–17]. However, daptomycin dosage adjustments for pediatric patients with renal impairment have not yet been established because such patients have been excluded from participation in previous clinical trials [12,18].

Regarding safety, daptomycin is listed among the key potentially inappropriate drugs in pediatric patients owing to the risk of neuromuscular and skeletal adverse events [19]. However, the incidence of risk factors for daptomycin-related adverse events in pediatric patients has not been investigated in detail. Unlike the recommendation that daptomycin

should be a first-line therapy for adult patients with bacteremia and cSSTI, the Practical Guidelines for the Management and Treatment of Infections caused by MRSA, 2019 Edition, recommend daptomycin as a second-line therapy for pediatric patients. This recommendation is made after observing persistent bacterial growth and aggravation of clinical symptoms in those receiving vancomycin therapy owing to the very limited experience with daptomycin in pediatric patients [20]. The proposed dosage of daptomycin for pediatric patients is based on the age; however, dosage adjustments for those with renal impairment have not been well-established, because such patients were not included in clinical trials or previous studies. Furthermore, although we previously investigated the association between age and the onset of daptomycin-induced adverse events using the U.S. food and drug administration adverse event reporting system (FAERS) [21], the FAERS database does not include clinical laboratory data, such as serum creatinine levels; thus, we could not evaluate renal functions as a risk factor for daptomycin-related adverse events. Additionally, while daptomycin dosage was stratified by age and sources of infection, 5,084 reports lacked information on infection sources. Consequently, we were unable to assess the impact of daptomycin dosage.

Therefore, this study aimed to investigate the clinical effectiveness of daptomycin and incidence of adverse events associated with its use in pediatric patients. Associations between clinical outcomes and risk factors, including daptomycin dosage and renal function, were also investigated.

## Materials and methods

### Ethics statement

This study was conducted in accordance with the Declaration of Helsinki and its amendments after obtaining approval from the Clinical Research Ethics Review Committee of Mie University Hospital (approval number H2023–078). The data were accessed for research purposes after May 2, 2023. Informed consent was obtained through an opt-out method from all participants because the data were collected retrospectively from electronic medical records. We accessed information that could identify individual participants after data collection. Data were collected between initiation of daptomycin and discontinuation of daptomycin or till December 2023.

### Patient population

We retrospectively reviewed the medical records of pediatric patients who received daptomycin therapy for bacteremia and cSSTI at Mie University Hospital, Aichi Medical University Hospital, Nagasaki University Hospital, Ryukyu University Hospital, and Nagoya University Hospital between September 1, 2011, and December 31, 2022. The following patients were excluded from the study: patients receiving daptomycin for <3 days, those aged <1 year, and those without bacteremia or cSSTI because the dosage is not specified in the package insert. We defined the following doses as approved doses according to the daptomycin package insert and renal impairment (S1 Table) [22]. Patients were classified based on the daptomycin dose according to the U.S. FDA-approved labeling (underdose, >1 mg/kg less than the approved dose; adequate dose, approved dose ±1 mg/kg; and overdose, >1 mg/kg more than the approved dose). Since daptomycin was administered based on body weight, we took into account a 1 kg range.

Obesity was defined as a body mass index (BMI) ≥30 kg/m$^2$, and for these patients, an adjusted body weight was used to calculate the daptomycin dose per body weight. Adjusted body weight was calculated as ideal body weight + 0.4 × (actual body weight − ideal body weight). This approach yields an AUC/MIC ratio that more closely correlates with that of individuals with normal weight [23].

### Data collection

At least 3 days prior to the initiation of daptomycin therapy, data on patient demographics, laboratory findings, hospitalization history, and sources of infection were collected using medical chart reviews. We extracted data on demographics (sex, age, and body weight), daptomycin therapy (daily dose and treatment duration), clinical laboratory data (serum albumin, blood urea nitrogen, serum creatinine, aspartate aminotransferase, alanine aminotransferase, hemoglobin,

C–reactive protein, creatine phosphokinase [CPK], white blood cells, and eosinophils), vital signs (body temperature, pulse rate, respiratory rate, and blood pressure), hemodialysis (HD), continuous hemodiafiltration (CHDF), and blood culture. We also extracted data on concomitant medications known to cause daptomycin-related CPK elevation (statin, fibrate, selective serotonin reuptake inhibitor, *β*-blocker, antipsychotics, colchicine, steroids, amiodarone, cyclosporine, propofol, and antihistamine) [24–25]. Renal function was classified using renal function criteria based on the estimated glomerular filtration rate (eGFR) [26]. The frequencies at which CPK values were measured were also recorded.

## Clinical effectiveness

Clinical effectiveness was determined by physicians at the end of daptomycin therapy and was categorized as follows: cure, improvement, failure, and non-evaluable. "Cure" was defined as the resolution of clinical signs and symptoms, with no indications for additional antibiotic therapy and/or a negative culture reported at the end of therapy. "Improvement" was defined as the partial resolution of clinical signs and symptoms and/or an indication for additional antibiotic therapy at the end of therapy. "Failure" was defined as persistent fever or worsening symptoms 48 hours after initiating therapy. "Non-evaluable" was defined as having insufficient information to determine the response at the end of treatment [18]. The clinical success rate was defined as the sum of cases cured and showed improvement at the end of daptomycin therapy [27]. In addition, mortality rates during daptomycin therapy were investigated.

## Microbiological effectiveness

Microbiological effectiveness was assessed by physicians at the end of daptomycin therapy and categorized as follows: initially negative, disappeared, non-evaluable, and microbiological failure. "Non-evaluable" was defined as having insufficient information to determine the response at the end of treatment. "Microbiological failure" was defined as the presence of continued positive blood cultures.

## Detection of adverse events

Daptomycin-related adverse events from the initiation of daptomycin administration to 30 days after the end of daptomycin administration were investigated. Laboratory tests were routinely conducted by physicians based on the patient's condition. Adverse events beyond biochemical disturbances (e.g., rashes, fevers, or gastrointestinal effects) were assessed through a review of medical charts. Adverse events deemed by physicians to be caused by daptomycin were analyzed.

## Statistical analyses

Statistical analyses were performed using JMP Pro 16 (SAS Institute, Cary, NC, USA). Categorical data are presented as numbers (percentages). Continuous variables are presented as medians with minimum and maximum values. The chi-square test or Fisher's exact test was used for categorical variables, and the Mann–Whitney U test was used for continuous variables. The clinical characteristics of the registered patients stratified by daptomycin dose were evaluated using the Kruskal–Wallis test. Statistical significance was defined as a two-tailed *P*-value <0.05.

## Results

### Patients

Fig 1 depicts the patient selection process. The following patients were excluded: those who received daptomycin for <3 days (n = 6), patients aged <1 year (n = 1), cases where dosage classification based on renal function and body weight was difficult to determine (n = 2), and those without bacteremia or cSSTI (n = 17). The infectious diseases other than bacteremia or cSSTI are detailed in S2 Table. A total of 54 patients (14 with cSSTI and 40 with bacteremia) were enrolled in this study. Daptomycin was administered to patients whose MIC were classified as susceptible. Bacteria and MIC values

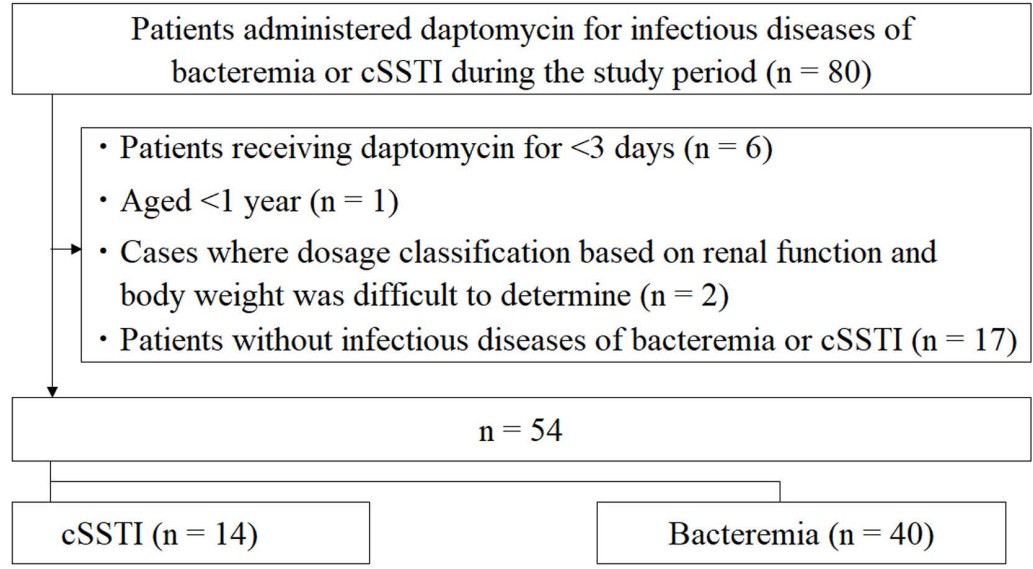

**Fig 1. Flowchart of the inclusion and exclusion of patients in this study.** cSSTI, complicated skin and skin tissue infection.

for patients with positive blood cultures are shown in S3 Table. No patients developed pneumonia. One case of bacteremia associated with a subcutaneous abscess was classified as cSSTI. Table 1 summarizes the baseline characteristics of the patients. The median (minimum–maximum) age and body weight were 16.0 years (1.0–18.0 years) and 44.2 kg (10.6–105.0 kg), respectively. One patient had a BMI > 30 (83.0 kg), and an adjusted body weight of 69.6 kg was used to calculate the daptomycin dose per body weight. The median daptomycin treatment duration was 15.5 days (3.0–106.0 days). Regarding the daptomycin dose, four (7%) patients received an underdose, 40 (74%) received an adequate dose, and 10 (19%) received an overdose. One patient (2%) underwent HD treatment, and four (7%) underwent CHDF therapy. Patients receiving dialysis therapy were excluded, and the number of patients with eGFR ≥ 90, 60–89, 30–59, 15–29, and <15 mL/min/1.73 m$^2$ were 42 (78%), 3 (6%), 3 (6%), 0 (0%), and 1 (2%), respectively. CPK values and eosinophil counts at the initiation of daptomycin therapy were 11.0 U/L (0.0–3,324.0 U/L) and 12.0/µL (0.0–1,120.0/µL), respectively.

Daptomycin was administered as a first-line treatment to 43 (80%) patients and as a second-line treatment to 11 (20%) patients because of adverse events or ineffectiveness of other anti-MRSA drugs. CPK values were measured every 4–7 days in 23 (43%) patients, every 2–3 days in 11 (20%) patients, every 10–14 days in nine (17%) patients, only once in three (6%) patients, and at >14–day intervals in one (2%) patient, while they were not measured during daptomycin therapy in seven (13%) patients (Fig 2). The age-stratified clinical characteristics of the registered patients are presented in S4 Table.

## Clinical and Microbiological effectiveness

Overall clinical success was achieved in 40 patients. Excluding 10 patients for whom clinical effectiveness was not evaluable, the clinical success rate reached 91%, with 24 (55%) classified as cured and 16 (36%) showing improvement (Fig 3). Three cases of treatment failure involved patients who received an underdose of daptomycin for bacteremia.

Microbiological effectiveness was categorized as initially negative (n = 30), disappeared (n = 16), non-evaluable (n = 5), or microbiological failure (n = 3) (Table 1). Microbiological failure was significantly higher in patients receiving an underdose of daptomycin ($P < 0.001$).

**Table 1.  Clinical characteristics of registered patients divided by daptomycin dose.**

| | Underdose (n = 4) | Adequate dose (n = 40) | Overdose (n = 10) | P–value |
|---|---|---|---|---|
| Age, years | 2.5 [2.0–11.0] | 16.0 [1.0–18.1] | 16.5 [4.0–18.0] | 0.035 |
| 1–<2 years, n (%) | 0 (0) | 2 (5) | 0 (0) | 0.020 |
| 2–6 years, n (%) | 3 (75) | 4 (10) | 1 (10) | |
| 7–11 years, n (%) | 1 (25) | 2 (5) | 1 (10) | |
| 12–17 years, n (%) | 0 (0) | 32 (80) | 8 (80) | |
| Male, n (%) | 1 (25) | 23 (58) | 9 (90) | 0.070 |
| Weight, kg | 17.1 [10.7–29.5] | 46.3 [10.6–105.0] | 45.8 [16.7–83.0] | 0.036 |
| Treatment duration, day | 20.0 [15.0–94.0] | 14.5 [3.0–106.0] | 20.0 [3.0–78.0] | 0.420 |
| Hemodialysis, n (%) | 0 (0) | 1 (3) | 0 (0) | 0.801 |
| Continuous hemodiafiltration, n (%) | 2 (50) | 1 (3) | 1 (10) | 0.002 |
| Renal function in patients without dialysis therapy, n (%) | 2 (50) | 38 (95) | 9 (90) | – |
| eGFR ≥ 90, mL/min/1.73 m$^2$, n (%) | 1 (25) | 32 (80) | 9 (90) | 0.014 |
| eGFR 60–89, mL/min/1.73 m$^2$, n (%) | 0 (0) | 3 (8) | 0 (0) | |
| eGFR 30–59, mL/min/1.73 m$^2$, n (%) | 1 (25) | 2 (5) | 0 (0) | |
| eGFR 15–29, mL/min/1.73 m$^2$, n (%) | 0 (0) | 0 (0) | 0 (0) | |
| eGFR < 15, mL/min/1.73 m$^2$, n (%) | 0 (0) | 1 (3) | 0 (0) | |
| Alb, g/dL | 3.7 [2.9–4.6] | 3.5 [1.3–4.7] | 3.2 [2.4–3.8] | 0.313 |
| BUN, mg/dL | 32.0 [7.2–70.3] | 12.9 [3.5–78.0] | 10.0 [7.0–25.0] | 0.358 |
| Scr, mg/dL | 0.4 [0.1–0.7] | 0.5 [0.1–7.1] | 0.5 [0.3–1.1] | 0.362 |
| AST, U/L | 67.5 [8.0–124.0] | 23.5 [8.0–173.0] | 19.0 [12.0–77.0] | 0.454 |
| ALT, U/L | 26.0 [11.0–43.0] | 25.5 [6.0–184.0] | 22.0 [7.0–154.0] | 0.871 |
| Hb, g/dL | 8.8 [7.1–13.4] | 9.4 [5.9–14.6] | 10.7 [6.9–13.3] | 0.301 |
| CRP, mg/dL | 3.8 [0.1–9.2] | 1.3 [0.0–27.1] | 5.1 [0.0–17.4] | 0.063 |
| Eosinophil count,/µL | – | 1.0 [0.0–1,120.0] | 130.0 [0.0–747.0] | 0.019 |
| CPK, U/L | 6.0 [4.0–86.0] | 4.0 [0.0–3,324.0] | 26.0 [0.0–558.0] | 0.149 |
| cSSTI, n (%) | 0 (0) | 8 (20) | 6 (60) | 0.088 |
| Bacteremia, n (%) | 4 (100) | 32 (80) | 4 (40) | |
| **Concomitant medications** | | | | |
| Statin, n (%) | 0 (0) | 0 (0) | 0 (0) | 1.000 |
| Fibrate, n (%) | 0 (0) | 3 (8) | 0 (0) | 0.501 |
| SSRI, n (%) | 0 (0) | 0 (0) | 0 (0) | 1.000 |
| β-blocker, n (%) | 1 (25) | 0 (0) | 1 (10) | 0.020 |
| Antipsychotics, n (%) | 0 (0) | 2 (5) | 0 (0) | 0.831 |
| Colchicine, n (%) | 0 (0) | 0 (0) | 0 (0) | 1.000 |
| Steroids, n (%) | 3 (75) | 8 (20) | 1 (10) | 0.036 |
| Amiodarone, n (%) | 0 (0) | 0 (0) | 0 (0) | 1.000 |
| Cyclosporine, n (%) | 0 (0) | 3 (8) | 0 (0) | 0.501 |
| Propofol, n (%) | 0 (0) | 3 (8) | 0 (0) | 0.501 |
| Antihistamine, n (%) | 0 (0) | 8 (20) | 1(10) | 0.326 |
| **Clinical effectiveness** | | | | |
| Cure, n (%) | 1 (25) | 17 (43) | 5 (50) | 0.746 |
| Improvement, n (%) | 0 (0) | 14 (35) | 2 (20) | 0.329 |
| Failure, n (%) | 3 (75) | 0 (0) | 1 (10) | <0.001 |
| Non-evaluable, n (%) | 0 (0) | 9 (23) | 2 (20) | 0.567 |
| Death, n (%) | 3 (75) | 0 (0) | 1 (10) | <0.001 |

*(Continued)*

**Table 1.** (Continued)

| | Underdose (n = 4) | Adequate dose (n = 40) | Overdose (n = 10) | P–value |
|---|---|---|---|---|
| **Microbiological effectiveness** | | | | |
| Initially negative, n (%) | 1 (25) | 23 (58) | 6 (60) | 0.437 |
| Disappear, n (%) | 0 (0) | 13 (33) | 3 (30) | 0.398 |
| Non-evaluable, n (%) | 0 (0) | 4 (10) | 1 (10) | 0.778 |
| Microbiological failure, n (%) | 3 (75) | 0 (0) | 0 (0) | <0.001 |

Alb, serum albumin; ALT, alanine aminotransferase; AST, aspartate transaminase; BUN, blood urea nitrogen; CPK, creatine phosphokinase; CRP, C–reactive protein; cSSTI, complicated skin and skin structure infection; eGFR, estimated glomerular filtration rate; Hb, haemoglobin; Scr, serum creatinine; SSRI, selective serotonin reuptake inhibitor.

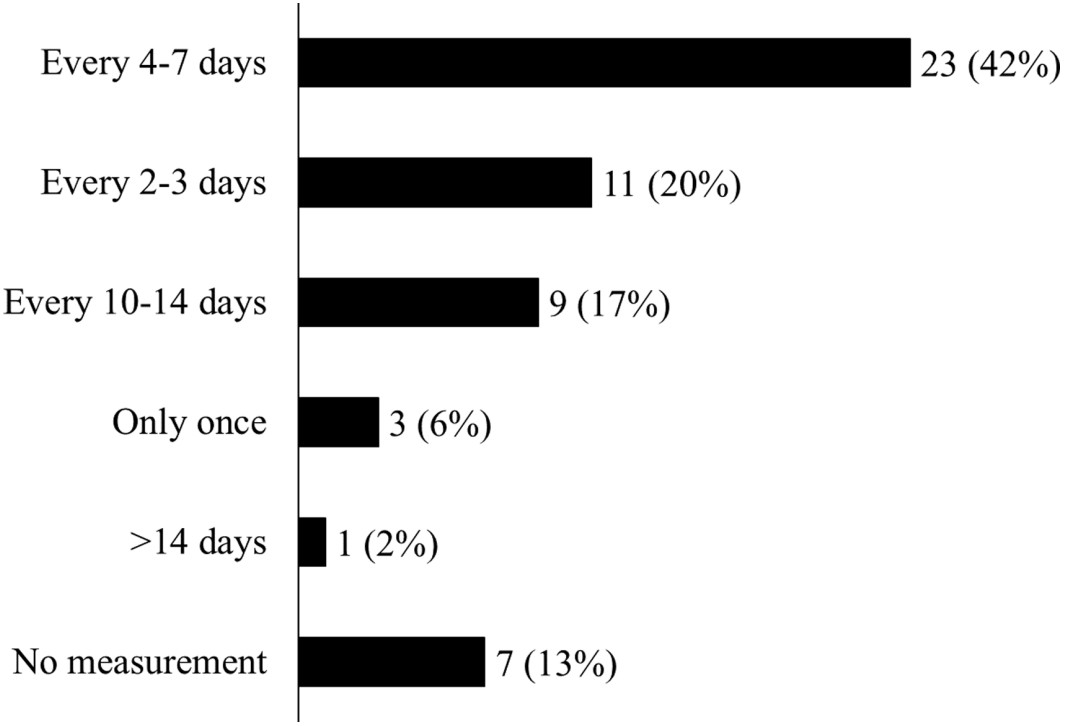

**Fig 2. Frequency of CPK measurements.** CPK, creatine phosphokinase.

Of the 54 patients, four (7%) died. The rates of microbiological failure and underdosing of daptomycin in these patients were significantly higher: microbiological failure was observed in two of four (50%) patients who died compared with one of 50 (2%) patients who survived (P = 0.012). Regarding dosing, three of the four patients who died (75%) were underdosed; none (0%) received an adequate dose; and one (25%) received an overdose. In contrast, among the surviving patients (n = 50), one (2%) was underdosed, 40 (80%) received an adequate dose, and nine (18%) received an overdose (P < 0.001).

Among the 11 patients who received daptomycin as a second-line treatment, two were infected with methicillin-resistant *coagulase-negative staphylococci*, one with MRSA, and three with methicillin-resistant *Staphylococcus epidermidis* (MRSE). The remaining patients included one with *Leuconostoc lactis*, one with *coagulase-negative staphylococci*,

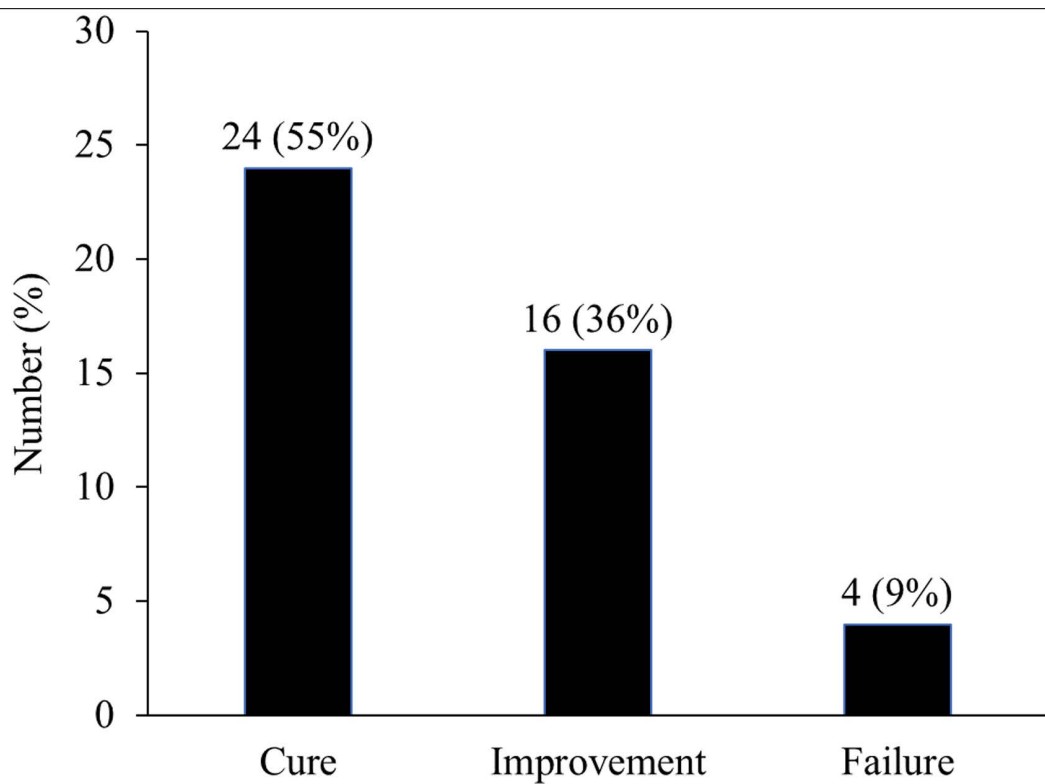

**Fig 3. Clinical effectiveness.**

and three with insufficient information. Among the six patients with antibiotic-resistant infections, one was cured, two improved, two showed partial improvement, and one was non-evaluable. Five of the six patients showed disappearance of infection, while one experienced microbiological failure.

### Adverse events

During the study period, seven (13%) patients experienced daptomycin-related adverse events: CPK elevation observed on days 9 (baseline, 8.0 U/L to day 9, 1,116 U/L) and 106 (baseline, 17.0 U/L to day 106, 2,013 U/L) in two patients, eosinophilic pneumonia observed on day 8 in one patient, rhabdomyolysis observed on day 3 in one patient, liver failure observed on days 6 and 15 in two patients, and drug fever observed on day 26 in one patient. Concerning drug fever, rifampicin was concurrently discontinued, rendering the identification of the contributing drug impossible. The drug lymphocyte stimulation test returned negative results for both drugs, yet the fever subsided following discontinuation of the drugs. Patients with liver failure did not exhibit elevated CPK levels. Table 2 presents the clinical features of the patients who experienced daptomycin-related adverse events. Among these patients, five were aged 12–18 years and two were aged 1 year. Five patients received an adequate dose of daptomycin, and two patients received an overdose. Six patients had an eGFR ≥90 mL/min/1.73 m$^2$, and one patient had an eGFR of 60–89 mL/min/1.73 m$^2$. In terms of clinical effectiveness at the end of daptomycin therapy, three patients were cured, three showed improvements, and one was non-evaluable. The clinical characteristics of patients with and without daptomycin-related adverse events are summarized in S5 Table. Adverse events were observed regardless of renal function.

**Table 2. Characteristics of patient's with daptomycin-related adverse events.**

| Case | Adverse events | Sex | Age, years | Weight, kg | Dose/kg per day, g/kg per day | Does | Administration period, day | Type of Infection | Bacteria |
|---|---|---|---|---|---|---|---|---|---|
| 1 | CPK elevation | F | 18 | 42.9 | 8.0 × 1 | Adequate dose | 9 | Bacteremia | MRCNS |
| 2 | CPK elevation | M | 12 | 39.7 | 4.0 × 1 | Adequate dose | 106 | cSSTI | – |
| 3 | Eosinophilic pneumonia | M | 16 | 49.9 | 6.0 × 1 | Adequate dose | 8 | Bacteremia | – |
| 4 | Rhabdomy-olysis | M | 17 | 83.0 | 10.1 × 1 | Overdose | 3 | Bacteremia | *Staphylococcus haemolyticus* |
| 5 | Liver failure | F | 1 | 11.3 | 11.0 × 1 | Adequate dose | 6 | Bacteremia | MRSE |
| 6 | Liver failure | M | 16 | 43.0 | 8.0 × 1 | Overdose | 15 | cSSTI | – |
| 7 | Drug fever | M | 1 | 15.0 | 9.0 × 1 | Adequate dose | 26 | Bacteremia | MSSA |

| Case | Scr, mg/dL | eGFR, mL/min/1.73 m² | AST, U/L | ALT, U/L | CPK, U/L | Eosinophil count,/µL | Concomitant medications | Clinical effectiveness | Change other anti–MRSA antibiotics |
|---|---|---|---|---|---|---|---|---|---|
| 1 | 0.8 | ≥90 | 17.0 | 12.0 | 8.0 | – | – | Cure | NA |
| 2 | 0.7 | ≥90 | 14.0 | 37.0 | 17.0 | 0.0 | Steroids | Improvement | NA |
| 3 | 0.7 | ≥90 | 51.0 | 112.0 | – | 0.0 | – | Improvement | LZD |
| 4 | 0.6 | ≥90 | 26.0 | 100.0 | 26.0 | 33.0 | Steroids | Non-evaluable | NA |
| 5 | 0.7 | ≥90 | 109.0 | 123.0 | 20.0 | – | – | Improvement | LZD |
| 6 | – | ≥90 | 12.0 | 10.0 | 20.0 | 580.0 | – | Cure | NA |
| 7 | 0.4 | 60–89 | 33.0 | 15.0 | 70.0 | – | Antihistamine | Cure | NA |

F, female; M, male; CPK, creatine phosphokinase; cSSTI, complicated skin and skin structure infection; MRCNS, methicillin-resistant coagulase-negative *Staphylococci*; MRSE, methicillin-resistant *Staphylococcus epidermidis*; MSSA, methicillin-susceptible *Staphylococcus aureus*

ALT, alanine aminotransferase; AST, aspartate; CPK, creatine phosphokinase; eGFR, estimated glomerular filtration rate; Scr, serum creatinine; LZD, linezolid; MRSA, methicillin-resistant *Staphylococcus aureus*; NA, not available.

## Discussion

This multicenter retrospective observational study evaluated the influence of renal function and daptomycin dosage on the clinical effectiveness of the drug and related adverse events in Japanese pediatric patients. By conducting a multicenter collaborative study, we believe that we were able to not only increase the sample size, but also exclude site-specific data. Excluding 15 patients in whom clinical effectiveness was not evaluable, the clinical success rate reached 91%. This rate was similar to or better than that reported by previous clinical trials [12,13,18]. These findings suggest that daptomycin is generally effective in treating bacteremia and cSSTI in Japanese pediatric patients. The underdose group had a significantly larger proportion of younger patients as well as a higher incidence of mortality and microbiological failure. During daptomycin therapy, seven (13%) patients experienced daptomycin-related adverse events, regardless of the daptomycin dosage and renal function.

Daptomycin exhibits exposure-dependent bactericidal activity. The approved doses for pediatric patients according to the package insert have yielded daptomycin exposures comparable with those previously reported in adults [6,28]. Moreover, each age-adjusted dosing regimen in pediatric patients resulted in daptomycin concentration levels similar to those observed in adult patients [6,28]. Renal blood flow increases in proportion to the development of renal tubules, reaching adult levels by the age of 5 months [29]. The bactericidal effect of daptomycin is generally associated with an AUC/MIC ≥200 [30–32], while a $C_{max}$/MIC ratio between 12 and 94 is considered optimal for achieving bacteriostatic effects [30] and

minimizing bacterial resistance [33]. Higher dosages of daptomycin are recommend for the treatment of MRSA bacteremia, and ineffectiveness in some patients may be attributed to suboptimal dosing [34]. In this study, four patients (7%) receiving an underdose of daptomycin required a high dose to treat bacteremia. Of these four patients, two were infected with MRSE, one with *Leuconostoc lactis*, and one with *Staphylococcus epidermidis.* The specific reason for the low-dose administration in this case is unclear; however, it is potentially related to the absence of MRSA infection. The high rates of microbiological failure and mortality observed in patients receiving an underdose of daptomycin highlight the importance of administering approved doses as specified in the package insert. Furthermore, a previous study revealed that the pediatric package label dosing of DAP resulted in insufficient drug exposure, with the probability of target attainment for efficacy reaching only 26.3-50.1% in pediatric patients compared to the desired goal of ≥90.0%. Based on pharmacokinetic/pharmacodynamic analysis, doses higher than those currently recommended in the pediatric labeling would be necessary to achieve validated efficacy targets [35]. Therefore, further studies are warranted to investigate the relationship between DAP concentration, dosage, and clinical effectiveness.

According to what has been reported in the literature, daptomycin is used for the treatment of serious infections caused by Gram-positive bacteria, especially for those strains that present resistance to the usual therapeutic options [35–37]. In this study, daptomycin was used as a second-line treatment in 11 patients due to adverse events or insufficient effectiveness from vancomycin or teicoplanin. Among these, two patients experienced adverse events but showed therapeutic effectiveness. Three patients who received an underdose unfortunately passed away, while six patients who received adequate dose or overdose demonstrated therapeutic effectiveness without adverse events. Due to the small sample size, the impact of prior treatments on daptomycin effectiveness could not be confirmed, but future studies will aim to gather more data.

The frequency of daptomycin-related adverse events was similar to that observed in clinical trials on adults (14%) and pediatric patients (11–15%) [12,13,18]. Our findings indicate that daptomycin is a safe once-daily agent for pediatric patients. However, unlike in previous studies involving adult patients, daptomycin-related adverse events developed in pediatric patients receiving the approved dose regardless of renal impairment [22]. Of the 54 patients, 10 (19%) received an overdose, but the occurrence of adverse events was not related to the daptomycin dose. Previous reports in adult patients have revealed that $C_{min}$ ≥24.3 mg/L is most significantly associated with CPK elevation. Additionally, factors such as total body weight >111 kg, prior or concomitant therapy with a $\beta$-hydroxy-$\beta$-methylglutaryl–coenzyme A reductase inhibitor, and renal impairment have been identified as risk factors for CPK elevation [38]. However, in Japanese pediatric patients, no apparent relationship was observed between CPK elevations and daptomycin exposure [39]. S4 Table presents the clinical characteristics of patients with and without daptomycin-related adverse events. No significant relationship was found between adverse events and patient characteristics, including body weight and renal function. Therefore, patients should be continuously monitored for adverse events during daptomycin administration.

The incidence rates of liver failure and CPK elevation in this study were similar to those observed in clinical trials involving adult and pediatric patients. Daptomycin-induced liver failure may onset between 2 and 14 days after the initiation of daptomycin therapy. Nonetheless, liver failure is readily reversible with the cessation of daptomycin therapy, without long-term sequelae [40]. In the present study, liver failure was observed 6 and 15 days after the initiation of daptomycin therapy. Daptomycin-related liver failure has been reported in adult patients without pre-existing liver failure [41–44]. Moreover, mild-to-moderate elevations in serum aminotransferase levels during daptomycin therapy have been demonstrated to be caused by idiosyncratic liver failure originating from the muscle rather than from the liver [45]. The liver injury observed in the present study was not accompanied by elevated CPK levels. Additionally, little evidence of an association between daptomycin and liver injury was observed. Since no reports have explored the risk factors for daptomycin-induced liver failure, further research is needed to elucidate the underlying mechanisms.

Monitoring CPK levels once a week or more frequently is recommended for adult patients [46]. In the present study, no CPK measurements were performed in seven (13%) patients throughout the daptomycin administration period.

Additionally, three patients (5%) underwent CPK measurements only at the initiation of daptomycin therapy. Two patients had elevated CPK levels on days 9 and 106 after daptomycin therapy was initiated. Preclinical studies have indicated that neonatal and juvenile dogs are more susceptible to the adverse muscular effects of daptomycin than adult animals [28]. Therefore, continuous monitoring of CPK levels is necessary to avoid discontinuation of daptomycin therapy [40].

Drug fever and suspected eosinophilic pneumonia were observed in this study. These have not been reported in pediatric patients in previous clinical trials. Antimicrobial agents are the most common causes of drug fever, accounting for approximately one-third of all reported cases [47]. The recurrence of fever in a patient who has defervesced on receiving antimicrobial treatment for infection may be misinterpreted as a relapse of the original infection. Daptomycin-induced eosinophilic pneumonia frequently develops among older adults [48–50]. Patients with signs and symptoms of eosinophilic pneumonia should immediately discontinue daptomycin treatment. Therefore, our findings suggest the need to monitor pediatric patients for drug fever and eosinophilic pneumonia.

This study has some limitations. First, this was a retrospective, observational study. Second, only 16 pediatric patients were younger than 11 years. However, owing to the multicenter nature of this study, the sample size was larger than that of a previous study on daptomycin use in children [51]. In addition, pediatric patients with renal impairment were included, unlike in previous clinical trials [18]. Third, an accurate assessment of severity of symptoms was not performed using the Common Terminology Criteria for Adverse Events grade. Fourth, we identified adverse events that clinicians believe may be associated with daptomycin, although causation could not be established with certainty. Fifth, although the classification of chronic kidney disease can be used as an indicator of renal function in pediatric patients, it cannot be used to rule out the possibility of acute kidney injury. Sixth, we did not measure the daptomycin concentration and categorized the initial dose as "appropriate" solely based on whether it matched the dose recommended in the package insert. Seventh, we could not investigate the association between the bactericidal effect of daptomycin and MIC/minimum bactericidal concentration (MBC). Further research, including information of daptomycin dose based on the MIC/MBC values depending on the bacterial species, is needed. Finally, since this was a multicenter retrospective observational study, antimicrobial stewardship programs varied across the different centers.

## Conclusions

This study demonstrated that an adequate dose of daptomycin improved clinical effectiveness and the mortality rate. However, adverse events were observed regardless of the daptomycin dosage and renal function. Continuous monitoring of adverse events should be conducted in patients receiving the approved dose as well as in those with normal renal function. Further investigations including daptomycin concentration are required to clarify this issue.

## Supporting information

**S1 Table. Recommended dose adjustments for daptomycin in pediatrics with renal impairment.**
(DOCX)

**S2 Table. Infectious diseases of excluded patients except for bacteremia or cSSTI (n = 17).**
(DOCX)

**S3 Table. Bacteria and MIC values for patients with positive blood cultures (n = 21).**
(DOCX)

**S4 Table. Age-stratified clinical characteristics of patients.**
(DOCX)

**S5 Table. Clinical characteristics of registered patients with and without daptomycin-related adverse events.**
(DOCX)

## Acknowledgments

We thank Editage (www.editage.com) for the English language editing.

## Author contributions

**Conceptualization:** Takayuki Inagaki, Satoshi Kakiuchi, Takeshi Tanaka.

**Data curation:** CHIHIRO SHIRAISHI, Jun Hirai, Naoya Nishiyama, Yasuhiro Miyagawa, Hiroyuki Honda.

**Formal analysis:** CHIHIRO SHIRAISHI.

**Funding acquisition:** CHIHIRO SHIRAISHI.

**Investigation:** CHIHIRO SHIRAISHI, Yasuhiro Miyagawa, Hiroyuki Honda.

**Methodology:** CHIHIRO SHIRAISHI, Hideo Kato.

**Project administration:** Hideo Kato, Jun Hirai.

**Software:** CHIHIRO SHIRAISHI.

**Supervision:** Hideo Kato, Jun Hirai, Naoya Nishiyama, Takayuki Inagaki, Nobuyuki Ashizawa, Hiroshige Mikamo, Takuya Iwamoto.

**Validation:** Naoya Nishiyama, Nobuyuki Ashizawa, Masashi Nakamatsu, Yasuhiro Miyagawa, Satoshi Kakiuchi, Ayumi Fujita, Masato Tashiro, Takeshi Tanaka, Koichi Izumikawa, Mao Hagihara, Nobuhiro Asai, Nobuaki Mori.

**Visualization:** CHIHIRO SHIRAISHI, Kazuko Yamamoto, Takeshi Tanaka, Koichi Izumikawa, Takuya Iwamoto.

**Writing – original draft:** CHIHIRO SHIRAISHI.

**Writing – review & editing:** Hideo Kato, Jun Hirai, Naoya Nishiyama, Takayuki Inagaki, Nobuyuki Ashizawa, Masashi Nakamatsu, Kazuko Yamamoto, Yasuhiro Miyagawa, Hiroyuki Honda, Satoshi Kakiuchi, Ayumi Fujita, Masato Tashiro, Takeshi Tanaka, Koichi Izumikawa, Mao Hagihara, Nobuhiro Asai, Nobuaki Mori, Hiroshige Mikamo, Takuya Iwamoto.

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
