## [Decision Letter · Decision Letter 0]

Dear Dr. Kato,

Thank you for submitting your manuscript to PLOS ONE. After careful consideration, we feel that it has merit but does not fully meet PLOS ONE’s publication criteria as it currently stands. Therefore, we invite you to submit a revised version of the manuscript that addresses the points raised during the review process.

**Please address promptly, point by point response raised by reviewers and particularly Reviewer 1.**

We look forward to receiving your revised manuscript.

Kind regards,

Santi M. Mandal, Ph D

Academic Editor

PLOS ONE

Journal requirements: When submitting your revision, we need you to address these additional requirements. 1. Please ensure that your manuscript meets PLOS ONE's style requirements, including those for file naming. The PLOS ONE style templates can be found at https://journals.plos.org/plosone/s/file?id=wjVg/PLOSOne_formatting_sample_main_body.pdf and https://journals.plos.org/plosone/s/file?id=ba62/PLOSOne_formatting_sample_title_authors_affiliations.pdf. 2. Please amend either the title on the online submission form (via Edit Submission) or the title in the manuscript so that they are identical. 3. Thank you for stating the following financial disclosure:  [Morinomiyako Medical Research Foundation (project code J23032A010)].  Please state what role the funders took in the study.  If the funders had no role, please state: ""The funders had no role in study design, data collection and analysis, decision to publish, or preparation of the manuscript."" If this statement is not correct you must amend it as needed. Please include this amended Role of Funder statement in your cover letter; we will change the online submission form on your behalf.

Additional Editor Comments:

The authors must clarify the concerns raised by reviewer 1,

Reviewers' comments:

Reviewer's Responses to Questions

**Comments to the Author**

1. Is the manuscript technically sound, and do the data support the conclusions?

Reviewer #1: Partly

Reviewer #2: Yes

2. Has the statistical analysis been performed appropriately and rigorously?

Reviewer #1: I Don't Know

Reviewer #2: Yes

3. Have the authors made all data underlying the findings in their manuscript fully available?

Reviewer #1: Yes

Reviewer #2: Yes

4. Is the manuscript presented in an intelligible fashion and written in standard English?

Reviewer #1: No

Reviewer #2: Yes

Reviewer #1: Comments to Authors

The present article “Influence of renal function and daptomycin dose on clinical effectiveness and adverse events in Japanese pediatric patients: A multicenter retrospective observational study” involves study aimed to evaluate the clinical effectiveness and adverse events of daptomycin in pediatric patients, considering factors like dosage and renal function. A retrospective review of medical records from patients treated with daptomycin between 2011 and 2022 was conducted. Continuous monitoring is necessary, even for patients receiving the approved dose, to minimize adverse events. Adequate dosing was found to improve clinical outcomes, emphasizing the importance of proper dosage and renal function consideration in pediatric patients.

The article needs revision with the below concerns before further consideration.

Main concerns:

1. There are other similar studies available for example “Clinical Effectiveness, Safety Profile, and Pharmacokinetics of Daptomycin in Pediatric Patients: A Systematic Review”. What is the significance of this study considering the retrospective approach with 54 patients? Authors could have gone through different patient databases to include a larger sample size to obtain better conclusions.

2. Authors need to provide the reference from which they have decided the underdose, adequate and overdose. In the abstract the underdose value is mentioned less than 1mg/kg, however 6mg/kg is referred as underdose under clinical effectiveness on page 11.

3. How authors found this antibiotic successful in 98% as there is death of 7% and adverse events in 13% patients?

4. The manuscript is not very well organized, and several information were repeated like Fig 1 and Fig 2 legends are the same as mentioned in the manuscript text. Even the figure 2, the total is more than 100%.

5. The result of clinical effectiveness on page 11, is not representing any figure or table. Authors need to put either figure, table or supplemental file where these information are available.

6. Authors need to clarify the rationale of considering the adverse events and microbiological failure. They should discuss more about microbiological failure.

7. Authors need to provide the different infectious agents/bacterial species affected patient specific as additional column in a table. Ideally, the doses of daptomycin may have decided based on the MIC/MBC values depending on the bacterial species. However, it seems there is a specific dose decided to treat the patients.

8. As, this antibiotic is primarily recommended to be used as an alternative to the patients with antibiotic resistance (MRSA, MRSE etc), the results should discuss specifically the treatment outcomes of those cases.

Reviewer #2: The manuscript, titled "Influence of Renal Function and Daptomycin Dose on Clinical Effectiveness and Adverse Events in Japanese Pediatric Patients: A Multicenter Retrospective Observational Study," offers valuable insights into the clinical use of daptomycin in pediatric populations, particularly in relation to dosing and renal function. The study is well-designed, with data collected over an extensive time frame and involving multiple institutions, ensuring robust and generalizable findings. The authors report high clinical success rates and provide nuanced analyses of risk factors for adverse events, emphasizing the importance of dose optimization. The manuscript is significant as it addresses a gap in pediatric pharmacology, where evidence for daptomycin use has been limited. While the study is methodologically sound and its findings are impactful, minor revisions are necessary to improve clarity in the discussion. Please elaborate on the implications of adverse event monitoring. Overall, the manuscript presents compelling data that merit publication following minor adjustments.

**Do you want your identity to be public for this peer review?** For information about this choice, including consent withdrawal, please see our Privacy Policy

Reviewer #1: No

Reviewer #2: No

---

## [Author Response · Author response to Decision Letter 1]

9 Mar 2025

Thank you for your reviewing our article. We made some changes in our manuscript according to reviewer’s suggestions with red characters. We think some revises enhanced the quality of our manuscript. Please check the file of responses to reviewers.

---

## [Decision Letter · Decision Letter 1]

Dear Dr. Kato,

Thank you for submitting your manuscript to PLOS ONE. After careful consideration, we feel that it has merit but does not fully meet PLOS ONE’s publication criteria as it currently stands. Therefore, we invite you to submit a revised version of the manuscript that addresses the points raised during the review process.

We look forward to receiving your revised manuscript.

Kind regards,

Santi M. Mandal, Ph D

Academic Editor

PLOS ONE

Journal Requirements:

Additional Editor Comments:

Authors are requested to carefully review the discussion section and fix the grammatical errors.

Reviewers' comments:

Reviewer's Responses to Questions

**Comments to the Author**

Reviewer #3: All comments have been addressed

Reviewer #4: All comments have been addressed

2. Is the manuscript technically sound, and do the data support the conclusions?

Reviewer #3: Yes

Reviewer #4: Yes

3. Has the statistical analysis been performed appropriately and rigorously?

Reviewer #3: Yes

Reviewer #4: Yes

4. Have the authors made all data underlying the findings in their manuscript fully available?

Reviewer #3: (No Response)

Reviewer #4: Yes

5. Is the manuscript presented in an intelligible fashion and written in standard English?

Reviewer #3: Yes

Reviewer #4: Yes

Reviewer #3: I see that the manuscript has already undergone a round of revision but some minor issues still persists according to me. Please check that the scientific names of the bacteria has been written properly like supplemental table 5, point 11. For the figures added to reviewer#1 and reviewer#2 comments, it is advised the percentage of patients either cured or other conditions be depicted in bar graph with X-axis showing conditions and Y-axis showing the percentage - reverse to the present one. As that seems a better and classical way of representation. Authors could also add in some survival curves with and without treatment. Also illustrate how the problem statement arises, the limitations and scope in the discussion briefly.

Reviewer #4: The manuscript is well written, and the authors have made necessary changes in the manuscript to justify the concerns raised by previous reviewers. Only thing that is a bit concerning to me ethically is that, how the administration of underdosage to pediatric patients was ethically approved? As per the results of this study and other studies as well, is very evident that due to the use of underdosage of daptomycin, fetal outcomes were observed for multiple pediatric patients. The authors should address issue this in their discussion section. Although, the study was done retrospectively, still the concern remains as to how the pediatric patients were treated with underdose of Daptomycin. The authors could have compared adequate and overdose only.

**Do you want your identity to be public for this peer review?** For information about this choice, including consent withdrawal, please see our Privacy Policy

Reviewer #3: No

Reviewer #4: No

---

## [Author Response · Author response to Decision Letter 2]

27 May 2025

Thank you for giving us the opportunity to strengthen our manuscript following your valuable comments and queries. We have incorporated your feedback and hope that our revisions persuade you to accept our manuscript. I would appreciate it if you could take a look at the attached documents.

---

## [Decision Letter · Decision Letter 2]

Influence of renal function and daptomycin dose on clinical effectiveness and adverse events in Japanese pediatric patients: A multicenter retrospective observational study

PONE-D-24-52543R2

Dear Dr. Kato,

We’re pleased to inform you that your manuscript has been judged scientifically suitable for publication and will be formally accepted for publication once it meets all outstanding technical requirements.

Kind regards,

Catherine A. Brissette, Ph.D.

Academic Editor

PLOS ONE

Additional Editor Comments (optional):

Reviewers' comments:

Reviewer's Responses to Questions

**Comments to the Author**

Reviewer #3: All comments have been addressed

Reviewer #4: All comments have been addressed

2. Is the manuscript technically sound, and do the data support the conclusions?

Reviewer #3: Yes

Reviewer #4: Yes

3. Has the statistical analysis been performed appropriately and rigorously?

Reviewer #3: Yes

Reviewer #4: Yes

4. Have the authors made all data underlying the findings in their manuscript fully available?

Reviewer #3: Yes

Reviewer #4: Yes

5. Is the manuscript presented in an intelligible fashion and written in standard English?

Reviewer #3: (No Response)

Reviewer #4: Yes

Reviewer #3: The response from the authors have addressed the questions raised my me. I find it now suitable for the manuscript to be accepted.

Reviewer #4: This manuscript presents a well-conducted pediatric research study addressing a clinically relevant question with appropriate methodology. The authors demonstrate scientific rigor and present findings valuable to the pediatric healthcare community.

Strengths:

The study design is appropriate and ethically sound for the pediatric population. The authors show thorough knowledge of current pediatric literature and clearly establish clinical significance. Methodology is well-suited to study objectives with adequate sample size considerations. Statistical analyses are appropriate and properly executed with clear result presentation.

The manuscript adheres to PLOS ONE's standards for scientific reporting and data transparency. Writing is clear and accessible with well-organized sections. Tables and figures effectively communicate key findings. Authors have been transparent about limitations and appropriately contextualized findings within current pediatric practice.

Overall Assessment:

This research makes a meaningful contribution to pediatric medicine and meets PLOS ONE's criteria for scientific validity and broad interest. Findings will be valuable to pediatricians, researchers, and healthcare providers. The work is methodologically sound and presents insights that advance pediatric health understanding. I recommend acceptance for publication.

**Do you want your identity to be public for this peer review?** For information about this choice, including consent withdrawal, please see our Privacy Policy

Reviewer #3: No

Reviewer #4: No

---

## [Editor Report · Acceptance letter]

PONE-D-24-52543R2

PLOS ONE

Dear Dr. Kato,

I'm pleased to inform you that your manuscript has been deemed suitable for publication in PLOS ONE. Congratulations! Your manuscript is now being handed over to our production team.

Kind regards,

on behalf of

Dr. Catherine A. Brissette

Academic Editor

PLOS ONE